# Repair Suggestions for Planning Domains with Missing Actions Effects

**Alba Gragera, Raquel Fuentetaja, Ángel García-Olaya, Fernando Fernández**

Universidad Carlos III de Madrid, Leganés, 28911, Madrid, Spain
agragera@pa.uc3m.es, {rfuentet, agolaya, ffernand}@inf.uc3m.es

### Abstract

In Automated Planning (AP) a proper definition of the domain and problem files is assumed. However, producing complete model descriptions is a time consuming and challenging task, especially for non-experts. It is easy to make mistakes when creating formal models, turning the planning task unsolvable for the planners. This can happen if the initial state is not fully and properly specified, some actions are missing, or some actions are incomplete. Explaining the absence of solution in such cases is essential to help humans in the development of AP tasks. In this paper we focus on repairing planning models where the effects of some actions are incomplete. We propose a compilation of the unsolvable task to a new extended planning task, where the actions are allowed to insert possible missing effects. The solution to such task is a plan that achieves the goals of the original problem while also warning about the modifications that were necessary to do so. Experimental results show this approach can be effectively used to repair incomplete planning tasks across different planning domains.

## 1 Introduction

Automated Planning (AP) is a general problem-solving technique for a wide range of scenarios and goals (Ghallab, Nau, and Traverso 2004). Planning tasks are usually defined by a domain description, which specifies all available actions and the predicates used to describe the states; and a problem description that contains the initial and goal states. Given a solvable and well-defined task, and assuming infinite memory and time resources, a planner will return a solution. However, there are some scenarios where neither completeness nor correctness in the planning task specification can be ensured (Kambhampati 2007). Flaws in the task model can appear due to a noisy acquisition process, because domain engineers are not experts in the description language or because they do not have deep knowledge of the current task, specially when the domain is difficult to represent. This can lead to a wrong or incomplete specification of the initial state, or an inaccurate actions' definition. Such issues can turn the planning task unsolvable: due to a loss of information, there is no way to achieve the goals from the initial state.

Previous works have considered initial states from which the goals cannot be achieved (Sreedharan et al. 2019; Göbelbecker et al. 2010). They provide explanations and different alternatives which would turn the task solvable, although they do not consider alterations in the domain and assume a proper specification. But just making changes to the initial state is not enough in some cases. Let us consider the well known barman domain (Linares López, Jiménez Celorrio, and García-Olaya 2015), where a robot prepares drinks using glasses that need to be empty and clean to be filled. Forgetting to include the action effect that cleans the glass (Figure 1) will turn the task unsolvable, but setting the glass as clean in the initial state will not solve the problem, since it will get dirty again. A better option would be to repair the operator and allow the robot to clean the glass.

```
(:action clean-shot
   :pareters (?s - shot ?b - beverage ?h1 ?h2 - hand)
   :precondition (and (holding ?h1 ?s) (handempty ?h2)
                      (empty ?s) (used ?s ?b))
   :effect (and (not (used ?s ?b)) (clean ?s)))
```

Figure 1: Action to clean a shot. If the positive effect *(clean ?s)* is removed, the planner is not able to find a plan.

Due to the number of potential changes to the set of operators, repairing faulty domains is not trivial. Previous works focusing on incomplete domains assume guidelines from the user side to supply the lack of knowledge (McCluskey, Richardson, and Simpson 2002; Simpson, Kitchin, and McCluskey 2007; Nguyen, Sreedharan, and Kambhampati 2017; Garland and Lesh 2002).

This paper introduces domain reparation procedure based on AP for cases where some action effects are missing, without receiving additional information to the unsolvable task. Since some of the common mistakes when modelling an AP task are more likely to return an erroneous or undesirable plan (which can be supervised by the user), in this work we consider unsolvable tasks, where no plan is provided by the planner. Specifically, we focus on the absence of positive action effects, which can seriously affect the task solvability. Given a domain with incomplete positive effects for some actions, we compile the unsolvable task into a new extended plannning task, providing operators to repair any action of

the underlying domain with possible missing positive effects. The solution is a plan that achieves the goals while also including warnings about the modifications made to repair the domain model.

In the remainder of the paper we present the background in AP, followed by the problem formulation in Section 3. The compilation of the extended planning task is detailed in Section 4. Section 5 contains some problems related to domain reparation identified in the course of this work. We propose a metric to address them, detailed in the same section. Sections 6 and 7 include the experiments conducted and a discussion about our approach. Finally, we discuss some related works and draw the main conclusions of our work.

## 2 Background

Automated Planning tasks define problems which solutions are sequences of actions, called a plans, that achieve the problem goals when applied to specific initial states. We use the first-order (lifted) planning formalism, where a classical planning task is a pair $\Pi = (D, I)$, where $D$ is the planning domain and $I$ defines a problem instance. A planning domain is a tuple $D = \langle \mathcal{H}, \mathcal{C}, \mathcal{P}, \mathcal{A} \rangle$; where $\mathcal{H}$ is a type hierarchy; $\mathcal{C}$ is a set of (domain) constants; $\mathcal{P}$ is a set of predicates defined by the predicate name, and the types of its arguments; and $\mathcal{A}$ is a set of *action schemas*. If $p(t) \in \mathcal{P}$ is an $n$-ary predicate, and $t = t_1, \ldots, t_n$ are either typed constants or typed free variables, then $p(t)$ is an *atom*. An atom is *grounded* if its arguments do not contain free variables. Action schemas $a \in \mathcal{A}$ are tuples $a = \langle name(a), par(a), pre(a), add(a), del(a), cost(a) \rangle$, defining the action name; the action parameters (a finite set of free variables); the preconditions, add and delete lists; and the action cost. $pre(a)$ is a set of atoms representing what must be true in a state to apply the action. $add(a)$ and $del(a)$ represent the changes produced in a state by the application of the action (added and deleted atoms, respectively). A problem instance is a tuple $I = \langle \mathcal{O}, \mathcal{I}, \mathcal{G} \rangle$, where $\mathcal{O}$ is a set of typed constants representing problem-specific objects; $\mathcal{I}$ is the set of ground atoms in the initial state; and finally, $\mathcal{G}$ is the set of ground atoms defining the goals.

Grounded actions $\underline{a}$ are obtained from action schemas $a$ by substituting the free variables in the action schema' parameters by constants in $\mathcal{O}$. A grounded action $\underline{a}$ is applicable in an state $s$ if $pre(\underline{a}) \subseteq s$. When a grounded action is applied to $s$ we obtain a successor state $s'$, defined as $s' = \{s \setminus del(\underline{a})\} \cup add(\underline{a})$. A plan $\pi$ is a sequence of grounded actions $\underline{a}_1, \ldots, \underline{a}_n$ such that each $\underline{a}_i$ is applicable to the state $s_{i-1}$ generated by applying $\underline{a}_1, \ldots, \underline{a}_{i-1}$ to $\mathcal{I}$; $\underline{a}_1$ is applicable in $\mathcal{I}$; and the consecutive application of all actions in the plan generates a state $s_n$ containing the goals, $\mathcal{G} \subseteq s_n$. The cost of a plan is defined as $cost(\pi) = \sum_{\underline{a}_i \in \pi} cost(\underline{a}_i)$. Optimal plans are those with minimum cost.

## 3 Problem Formulation

We assume the previous lifted formalism to represent planning tasks, where planners usually rely on correctly and fully specified models. Incompleteness or flaws in these models

may result in the unsolvability of the task. And thus, the design of methods to repair automatically those incomplete planning models is a challenging research direction. This work aims to address such issues considering domain models with incomplete action definitions, giving rise to planning tasks where some necessary actions to achieve the goals do not generate all required positive effects, and these effects are also not generated by any other action, turning the task unsolvable. We define such domains as *add-incomplete*. Formally:

**Definition 1** (*Add-incomplete* planning domain). *A planning domain* $D^- = \langle \mathcal{H}, \mathcal{C}, \mathcal{P}, \mathcal{A}^- \rangle$ *is add-incomplete wrt. an underlying planning domain* $D = \langle \mathcal{H}, \mathcal{C}, \mathcal{P}, \mathcal{A} \rangle$ *iff:*

- *They define the same actions schemas with the same parameters (i.e. there is a one-to-one correspondence between action schemas);*
- *For every pair of corresponding action schemas,* $a \in \mathcal{A}$ *and* $a^- \in \mathcal{A}^-$:

$$a^- = \langle name(a), par(a), pre(a), add^-(a), del(a) \rangle$$

*where* $add^-(a) \subseteq add(a)$; *and*

- *There is at least one action schema in* $\mathcal{A}^-$ *for which* $add^-(a) \subset add(a)$.

For an *add-incomplete* domain there always exists at least one $a^- \in \mathcal{A}^-$ whose positive effects are a proper subset of the original action $a \in \mathcal{A}$. We will focus just on the cases where the missing effects turn the task unsolvable. In any other case, the found plan may not be semantically correct, but this might be more difficult to be detected automatically. For this work, we consider unitary actions for the underlying domain. Following the barman running example, an add-incomplete planning task could have the missing effect *(clean ?s)* for the action *clean-shot*. Since the glass has to be clean to prepare the beverage and no other actions can clean it, the task has no solution. Considering that different flaws can be spread over all the domain, other actions may also have missing positive effects. The solution consists in repairing the actions of the add-incomplete domain with additional add effects, so that the resulting planning task is solvable. Formally:

**Definition 2** (Repairing set). *Given an add-incomplete planning domain* $D^- = \langle \mathcal{H}, \mathcal{C}, \mathcal{P}, \mathcal{A}^- \rangle$, *with actions schemas* $a_i^- \in \mathcal{A}^-$, *a repairing set* $R = \{R_i\}_{i=1,\ldots,|\mathcal{A}^-|}$, *defines collections of atoms* $R_i = \{p(t) \mid p \subseteq \mathcal{P}, t \subseteq par(a_i^-)\}$ *that extend the add effects of every action* $a_i^-$, *resulting in a new (repaired) domain* $D^R = \langle \mathcal{T}, \mathcal{C}, \mathcal{P}, \mathcal{A}^R \rangle$ *with* $add(a_i^R) = add(a_i^-) \cup R_i$.[1]

We will denote the repaired domain as $D^R = D^- \oplus R$. The repairing set $R$ extends the action effects with new positive effects, provided that the terms in $t$ for every $p(t)$ also appear in the parameters of the action being repaired. Assuming that the domain of the underlying task corresponds the correct mental model in the mind of the designer, we are interested in finding the precise repairing set that extends the

---

[1]Note that the repairing set can be empty for some actions.

incomplete domain, generating exactly the underlying mental domain. We denote it as the true repairing set, defined as follows:

**Definition 3** (True repairing set). *Given an add-incomplete planning domain $D^-$ wrt. another planning domain $D$, the true repairing set, $R^*$, generates a repaired domain $D^{R^*} = D^- \oplus R^*$ which is exactly $D$, $D^{R^*} = D$.*

This is trivial if $D$ is known. However, when repairing $D^-$ we have no information about the underlying domain and how to fix it, which may lead to estimated reparations. Considering the barman domain again and according to that, a solution may not fix the action *clean-shot* with the *clean* effect, linking the predicate to another action instead. But we consider it a solution of the problem since it makes the task solvable. We formally define the problem to solve as:

**Definition 4** (Uninformed repairing problem). *Given an* **unsolvable** *planning task* $\Pi^- = (D^-, I)$ *with an add-incomplete domain wrt. the* **unkown domain** *of an* **assumed solvable** *planning task* $\Pi = (D, I)$, *the uninformed repairing problem consists of determining a repairing set $\hat{R}$ so that the planning task* $\Pi^{\hat{R}} = (D^{\hat{R}}, I)$, *with* $D^{\hat{R}} = D^- \oplus \hat{R}$, *is solvable.*

The objective of the uninformed repairing problem is to find the missing effects $\hat{R}$ that allow to extend $\Pi^-$ in a way that the resulting task, $\Pi^{\hat{R}}$, is solvable, ideally getting the repair to match the resulting domain with the domain of the underlying task. It is important to highlight that $\Pi$ and $R$ are unknown. Our approach to solve the problem is the topic of the next section.

## 4 Compilation to Classical Planning

In order to solve the uninformed repairing problem we perform a compilation of $\Pi^-$ to a new extended planning task $\Pi'$, provided with operators to repair any of the domain actions. At the conceptual level, every action scheme is now divided into three different parts: (1) the action in the original domain, (2) the reparation of the action and (3) the end of the reparation, where the current action is closed. Considering the running example, the planner first instantiates the *clean-shot* action. Since it is incomplete and it can not continue without actually cleaning the glass, it will fix it by applying a repair operator and, if no further repairs are necessary for that action, it will close such phase and continue with other actions. This process is repeated until the goals are achieved. If an action does not need to be repaired, it is closed without reparation.

To manage such reparation we need to have access to the planning task elements. First, the original type hierarchy is extended with new types yielding to a new type hierarchy $\mathcal{H}'$, shown in Figure 2. Types *action* and *type* (denoted in the remainder of the paper as $o\_act$ and $o\_type$, respectively) are used to provide information about the different action and type names present in the original domain $D^-$. *item* (denoted as $o\_item$) represents original domain objects and predicate names which are both elements required to repair the planning task. We use *object_domain* (denoted as $o\_obj$)

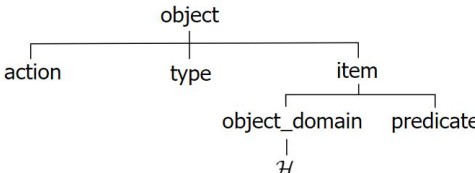

Figure 2: Hierarchy type proposed for the new extended planning task.

as an abstract object for summarizing the type hierarchy in the original domain.

The following new domain constants are introduced:

- Constants of type $o\_act$ to represent action names:
$$\mathcal{C}_a = \{name(a) \mid a \in \mathcal{A}\}$$

- Constants of type $o\_type$ to represent the types in the original hierarchy:
$$\mathcal{C}_t = \{t_{type} \mid type \in \mathcal{H}\}$$

- Constants of type $o\_pred$ to represent the names of the predicates in the original domain.
$$\mathcal{C}_p = \{p \mid p(t) \in \mathcal{P}\}$$

Examples of them can be seen in Figures 3 and 4. The former contains the types redefinition according to the presented hierarchy, color highlighting the new ones introduced in the compilation, whereas the second shows how the types *action*, *type* and *predicate* are instantiated through domain constants.

```
(:types
  action type item - object
  predicate objectdomain - item
  hand level beverage dispenser container - objectdomain
  ingredient cocktail  - beverage
  shot shaker  - container)
```

Figure 3: Reformulated types definition in $\Pi'$.

```
(:constants
    grasp leave fill-shot clean-shot ... - action
    handempty empty clean used shaked ... - predicate
    t_shot t_ingredient t_shaker ... - type
...)
```

Figure 4: Actions, predicates and types reformulated as domain constants in $\Pi'$.

Regarding predicates, we introduced two groups of new predicates in order to handle such reformulation. The first group contains predicates to access the elements of the original task; and the second group contains control predicates for the repairing task. The definition for the new *access predicates*, denoted as $\mathcal{P}_{access}$, is the following:

- **functor**(**o_pred**), that allows to define facts to represent predicate names. Possible examples are `functor(ontable)`, `functor(holding)`, etc.
- **type**(**o_obj**, **o_type**), that allows to define facts to access the type and super-types of every object in the original domain. Examples are `(type shaker1 t_shaker)` and `(type shaker1 t_container)`.
- **pred_⟨n⟩**(**o_pred**, **o_type₁**, ..., **o_typeₙ**), representing that there is a n-ary predicate defined in the original domain. It allows to define facts that facilitate the access to the predicates and the types of their arguments, as `(pred_2 contains t_container t_beverage)`. This means that there is a predicate of arity 2 to represent that a container contains a beverage.
- **in_state_⟨n⟩**(**o_pred**, **o_obj₁**, ..., **o_objₙ**), representing that the fact $(o\_pred, o\_obj_1, \ldots, o\_obj_n)$ is true in a state. Again, $n$ represents the arity. In this way we can group propositions with the same arity. Let us consider the predicates of the barman domain `(clean shot01)` and `(empty shot01)`, with arity 1, where `clean` and `empty` are predicate symbols and `shot01` is a object. They can be represented by the same predicate $in\_state\_1$. Then, if the shot is clean or empty in the current step our new task will not represent the grounded proposition `(clean shot01)`, but `(in_state_1 clean shot01)` or `(in_state_1 empty shot01)`. This lets us generalize our compiled task for any proposition in the domain, allowing the repair operators to add or remove any fact from the current state.
- **goal_⟨n⟩**(**o_pred**, **o_obj₁**, ..., **o_objₙ**) represents that $(o\_pred, o\_obj_1, \ldots, o\_obj_n)$ is a goal of the current planning task. An example would be `(goal_2 contains shot1 cocktail1)`.
- **add_eff**(**o_pred**, **o_act**), denotes that a predicate name appears in the add effects of an action. For instance, `(add_eff holding grasp)`.
- **del_eff**(**o_pred**, **o_act**), denotes that a predicate name appears in the del effects of an action. `(del_eff holding leave)`.

The new *control predicates*, denoted as $\mathcal{P}_{control}$, are:

- **checked**(**o_act**), denoting that an action has been already added to the plan, repaired or not. No action previously checked can be repaired.
- **current_action**(**o_act**), used to control what is the current action in course, to reparair it if needed.
- **patched**(**o_act**), indicating that action has been fixed at least once.
- **fix**(**o_act**, **o_pred**), meaning that an action has been repaired and the predicate involved.
- **used**(**o_item**), for repairing the action with the objects currently in use.
- **open**, indicating that the reparation is allowed.
- **fixed**, denoting that an action has been fixed and at least one of the add effects involved in a fix has been added to the current state.

```
(:action clean-shot
   :parameters (?s - shot ?b - beverage ?h1 ?h2 - hand)
   :precondition (and (in_state2 holding ?h1 ?s)
                      (in_state1 handempty ?h2)
                      (in_state1 empty ?s)
                      (in_state2 used ?s ?b)
                      (not (open)))
   :effect (and (not (in_state2 used ?s ?b))
                (current_action clean-shot)
                (used ?s) (used ?b)
                (used ?h1) (used ?h2)
                (open)))
```

Figure 5: Clean shot action compiled for the reparation task. It is important to note that the positive effect *clean ?s* is omitted and needs to be added by the planning process.

- **end**, used to mark the end of the whole reparation process.

Once this information has been presented, our new extended planning task can be defined as $\Pi' = (D', I')$, where $D' = \langle \mathcal{H}', \mathcal{C}', \mathcal{P}', \mathcal{A}' \rangle$. $\mathcal{H}'$ is the described new hierarchy (see Figure 2); the constants include the original constants and the new ones: $\mathcal{C}' = \mathcal{C} \cup \mathcal{C}_a \cup \mathcal{C}_t \cup \mathcal{C}_p$; the predicate definition includes the new access and control predicates: $\mathcal{P}' = \mathcal{P}_{access} \cup \mathcal{P}_{control}$; and $\mathcal{A}'$ is new set of actions schemes, defined as $\mathcal{A}' = \mathcal{A}_\alpha \cup \mathcal{A}_\varphi \cup \mathcal{A}_\psi \cup \mathcal{A}_\omega$, where $\mathcal{A}_\alpha$ are actions generated from the original domain actions, but compiled to adapt them to the new object representation; and $\mathcal{A}_\varphi$, $\mathcal{A}_\psi$ and $\mathcal{A}_\omega$ are the actions for the reparation. We describe all of them, as well as the configuration of the new problem instance $I'$, in the next paragraphs.[2]

**ACTIONS FROM ORIGINAL ACTIONS ($\mathcal{A}_\alpha$)** There is an action $\alpha_a \in \mathcal{A}_\alpha$ for every action $a \in \mathcal{A}$, with the same name and parameters, defined as follows.

$$
\begin{aligned}
name(\alpha_a) =& name(a) \\
par(\alpha_a) =& par(a) \\
pre(\alpha_a) =& \{in\_state\_\langle n \rangle(p, t) \mid p(t) \in pre(a)\} \cup \\
& \{\neg open\} \cup \{\neg end\} \\
add(\alpha_a) =& \{in\_state\_\langle n \rangle(p, t) \mid p(t) \in add(a)\} \cup \\
& \{current\_action(a)\} \cup \\
& \{used(x) \mid x \in par(a)\} \cup \{open\} \\
del(\alpha_a) =& \{in\_state\_\langle n \rangle(p, t) \mid p(t) \in del(a)\} \\
cost(\alpha_a) =& 0
\end{aligned}
$$

where $p(t)$, $t \subseteq par(a) \cup \mathcal{C}$, denotes any literal in the action preconditions or effects.

An example of the resulting compiled action is shown in Figure 5. Every predicate is now replaced by its *in-state* reformulation, and effects include information about what is

---

[2] The extended planning task requires in some cases of negative preconditions and forall effects. So, in this part we consider classical planning extended with these additional expresivity incorporated in ADL (Pednault 1989).

the current action been in course and also the objects involved in such action. Finally, we also include a flag fact called *open* to denote that, once applied, the action is allowed to be repaired. As long as this fact is present in the state, no other action in $\mathcal{A}_\alpha$ can be applied.

**FIX ACTION ($\mathcal{A}_\varphi$)** This action select any predicate and link it as a new effect of any action. The parameters are the variables $a$ and $p$, of type $o\_act$ and $o\_pred$, respectively. It follows the next scheme:

$$
\begin{aligned}
name(\varphi) =&\, fix \\
par(\varphi) =&\, (a, p) \\
pre(\varphi) =&\, \{current\_action(a)\} \cup \{functor(p)\} \cup \\
&\, \{\neg check(a)\} \cup \{open\} \\
add(\varphi) =&\, \{fix(a, p)\} \cup \{patched(a)\} \\
del(\varphi) =&\, \emptyset \\
cost(\varphi) =&\, C_\varphi
\end{aligned}
$$

The preconditions require to have an action opened to be repaired, that has not been previously checked (added to the plan). Actions have to be repaired in their first use, otherwise they can not be linked to a new effect after that. Then, a predicate symbol $p \in \mathcal{C}_t$ is selected to be attached as a new effect of the action, adding to the state that such action has to be patched. It has a constant associated cost which allows to define a metric for minimizing the number of reparations made in the domain.

**ADD-FIX ACTIONS ($\mathcal{A}_\psi$)** Once the missing effect has been linked to an action, such predicate symbol has to be matched with its parameters to be added to the state with the proper objects. The number of these objects depends on the arity $n$ of the predicate being added, so a predicate *clean* will be repaired with a single object of type *shot*, whereas other predicates may involve a larger number of objects. Accordingly, we need actions with different number of parameters to do so. For each arity $n$ we define a repair action $\psi_n$. The parameters are: variables $a$ and $p$ of type $o\_act$ and $o\_pred$, respectively; $n$ variables, $x_1, \ldots, x_n$, of type $o\_obj$, representing domain objects; and $n$ variables, $y_1, \ldots, y_n$ of type $o\_type$, representing their types. We define add-fix actions such that:

$$
\begin{aligned}
name(\psi_n) =&\, add\_fix\_\langle n \rangle \\
par(\psi_n) =&\, (a, p, x_1, \ldots, x_n, y_1, \ldots, y_n) \\
pre(\psi_n) =&\, \{current\_action(a)\} \cup \\
&\, \{fix(a, p)\} \cup \{functor(p)\} \cup \\
&\, \{pred\_\langle n \rangle(p, y_1, \ldots, y_n)\} \cup \\
&\, \{type(x_i, y_i) \mid 1 \le i \le n\} \cup \\
&\, \{used(x_i) \mid 1 \le i \le n\} \cup \{\neg end\} \\
add(\psi_n) =&\, \{in\_state\_\langle n \rangle(p, x_1, \ldots, x_n)\} \cup \{fixed\} \\
del(\psi_n) =&\, \emptyset \\
cost(\psi_n) =&\, 0
\end{aligned}
$$

**CLOSE ACTION ($\mathcal{A}_\omega$)** The application of the close action concludes the reparation of an action and change to the next

one. It deletes the *open* flag and the current action along with all objects used, adding the action as already checked to the current state. The scheme is defined as follows, where the only parameter is $a$ of type $o\_act$. Note that close actions have a *forall* effect, where $x$ is of type $o\_item$.

$$
\begin{aligned}
name(\omega) =&\, close \\
par(\omega) =&\, (a) \\
pre(\omega) =&\, \{current\_action(a)\} \cup \{open\} \\
add(\omega) =&\, \{checked(a)\} \\
del(\omega) =&\, \{\texttt{forall}(x, used(x))\} \cup \\
&\, \{current\_action(a)\} \cup \{open\} \\
cost(\omega) =&\, 0
\end{aligned}
$$

In summary, the new action schemes $\mathcal{A}' = \mathcal{A}_\alpha \cup \mathcal{A}_\varphi \cup \mathcal{A}_\psi \cup \mathcal{A}_\omega$, are defined as $\mathcal{A}_\alpha = \{\alpha_a \mid a \in \mathcal{A}\}$, with an action for every action in the original domain; $\mathcal{A}_\varphi = \{\varphi\}$, the fix action scheme; the add-fix action schemes $\mathcal{A}_\psi = \{\psi_n \mid n \, arity \, of \, p(t) \in \mathcal{P}\}$, with an action scheme for each different arity of the predicates in the original domain; and the close action scheme $\mathcal{A}_\omega = \{\omega\}$.

**PROBLEM INSTANCE ($I'$)** We have defined the compiled domain $D$ of the new planning task $\Pi' = (D', I')$. Now, we specify the new problem instance $I' = \langle \mathcal{O}, \mathcal{I}', \mathcal{G}' \rangle$. The new initial state contains all the facts present in the original one $I$, but reformulated according to the new type hierarchy (Figure 2) and the new predicate definition $\mathcal{P}'$. Then, predicates in $\mathcal{P}_{access}$ are instantiated in the new initial state. All of them represent static domain information about the original predicates and actions, except for $in\_state\_\langle n \rangle$ atoms. They are grounded with the available objects in $\mathcal{O}$ according to the facts currently true in the initial state. The initial state is then defined as follows, where the function $all(\mathcal{H}, o)$ provides all types (primitive type and all super-types) of an object in a type hierarchy:

$$
\begin{aligned}
\mathcal{I}' =&\, \{in\_state\_\langle n \rangle(p, t) \mid p(t) \in \mathcal{I}\} \cup \\
&\, \{functor(p) \mid p \in \mathcal{P}\} \cup \\
&\, \{pred\_\langle n \rangle(p, t_{type_1}, \ldots, t_{type_n}) \mid p \in \mathcal{P}\} \cup \\
&\, \{type(o, t_{type}) \mid o \in \mathcal{O}, t_{type} \in all(\mathcal{H}, o)\} \cup \\
&\, \{goal\_\langle n \rangle(p, t) \mid p(t) \in \mathcal{G}\} \cup \\
&\, \{add\_eff(p, name(a)) \mid p(t) \in add(a), a \in \mathcal{A}^-\} \cup \\
&\, \{del\_eff(p, name(a)) \mid p(t) \in del(a), a \in \mathcal{A}^-\}
\end{aligned}
$$

The new problem goals also contain the $in\_state\_\langle n \rangle$ facts corresponding to the facts defined in the goals of the original problem and additional *checked* facts for every action scheme, meaning that all action schemes have been considered to be repaired:

$$
\begin{aligned}
\mathcal{G}' =&\, \{in\_state\_n(p, t) \mid p(t)\} \in \mathcal{G}\} \cup \\
&\, \{checked(a) \mid a \in \mathcal{A}^-\}
\end{aligned}
$$

Actions become checked by the corresponding *close* action, when they are added to the plan, so we are indirectly forcing the planner to include all the domain actions in the plan. To alleviate this strong constraint we transform those

```
(:init
    (in_state1 clean shot01)
    (in_state1 empty shot01)
    (in_state2 dispenses dispenser01 ingredient01)
    (del_eff used clean-shot)
    ...)
(:goal (in_state2 contains shot01 cocktail04)
       (check clean-shot)
       ...)
```

Figure 6: Example of the problem compilation for the barman domain.

hard goals to soft goals following the compilation proposed by Keyder and Geffner (2009), including a *forgo* action that can directly achieve goals, but at high cost.

Figure 6 shows part of a barmans' problem instance as compiled in $I'$. We reformulate facts and extend the problem with information on already existing actions' effects, an input useful to guide the reparation process.

Given the set of actions defined, we establish the total cost of a solution as the sum of the reparations made in the domain (number of predicated linked as new effects). In the problem instance we include a metric to minimize such total cost. In this way we aim to solve the uninformed repairing problem to find the minimum repairing set $\hat{R}$ so that $D^- \oplus \hat{R}$, $\Pi^-$ is solvable.

## 5   Heuristics for Domain Reparation

There are a huge number of ways in which a planning domain can be repaired to make the task solvable, leading the process to fall into undesirable reparations. The proposed metric is established to minimize the number of reparations, but, in general, we have identified several problems related to the planning task reparation:

- **Goals**: a solution with lower cost and the simplest reparation may be to add the goal predicates to any of the actions, making the problem solvable by applying just such operator.
- **Adding deleted atoms**: including a new add effect that coincides with a del effect of the same action may not make many sense.
- **Repair effects in the same action**: the planner is likely to incorporate all required missing positive effects in the same action, without considering the rest of the planning actions.
- **Use of repaired actions**: it may be convenient for the planner the reparation of an action which is repeated throughout the plan, so we penalize the use of fixed actions in order not to overuse them.

Managing the reparations means to impose restrictions over the set of fixing operators, where the application of some of them may be penalized. To this end we decompose such actions into separate ones with different sets of preconditions and different costs, so that the least restrictive

fix actions have a higher cost. We established an optimization criteria which minimizes the total cost of the reparations made over the domain. Thus, the use of an optimal planner for solving the task guarantees a least costly solution. In the remainder of the section we show the different solutions considered to solve the mentioned issues.

**Goals**   We have defined an add-fix action with zero cost, even if the predicate being repaired is present in $\mathcal{G}$. However, to avoid achieving the solvability of the problem by repairing an action with predicates in the set of goals, we duplicate every add-fix action with arity $n$, $\psi_n$ for the case where the proposition being repaired is a goal, but penalizing its application. We define such actions as $\psi_n^g$. The parameters and effects are exactly those of the corresponding $\psi_n$ action. $\psi_n^g$ differs on the name, preconditions and costs:

$$name(\psi_n^g) = name(\psi_n)\_goal$$
$$pre(\psi_n^g) = pre(\psi_n) \cup \{goal\_\langle n \rangle (p, x_1, \ldots, x_n)\}$$
$$cost(\psi_n^g) = C_g$$

Conversely, common add-fix actions specify in the preconditions that the atom involved is not a goal. Considering the barman domain, if there is a goal *(contains cocktail01 shot01)*, a plan that repairs the *clean-shot* action and then prepares the cocktail will have a lower cost than a plan adding directly the goals. The cost $C_g$ is a constant value.

**Adding deleted atoms**   If we consider the action that fills the shot, it already involves a `(not (clean ?s))` atom as effect, so adding the same predicate as a positive effect would be an undesirable reparation. For this reason we include information about current action effects in the problem of the extended task, as shown in Figure 6. We take this into account in fix actions. We consider two types of fix actions here, $\varphi^-$ and $\varphi^+$, so that the former does not allow this kind of reparation. Both share the defined general scheme for fix actions ($\varphi$) in parameters and effects, where $\varphi^-$ is specified as follows:

$$pre(\varphi^-) = pre(\varphi) \cup \{\neg del\_eff(p, a)\}$$
$$cost(\varphi^-) = C^-$$

A $\varphi^-$ action can only be applied when the predicate being repaired is not part of the delete effects of the action. However, we have to penalize the opposite case instead of forbidding it, to allow for reparations adding atoms which predicate name is shared with a deleted atom. Remember that in our representation we only consider the predicate name in the $del\_eff$ facts, and there are domains with actions for which atoms sharing the same predicate name have to be added and deleted. For instance, in domains where the location of objects is important, it is necessary to delete the previous position and generate the new one. Thus, $\varphi^+$ actions represent this opposite case:

$$pre(\varphi^+) = pre(\varphi) \cup \{del\_eff(p, a)\}$$
$$cost(\varphi^+) = C^+$$

Actions $\varphi^-$ are prioritized over actions $\varphi^+$ making $C^+ > C^-$.

**Repair effects in the same action**   In order to avoid the planner adding all the missing predicates to the same action and to promote the use of the rest of the actions, the set of fix actions are specialized even more to cover the combination of cases considering delete effects and whether the action has (or has not) been already fixed in some way. This gives rise to the following four fix actions, which are now fully specified:

- Actions $\varphi_0^-$: the atom used to repair the action is not a negated effect and the action was not repaired before:

$$name(\varphi_0^-) = fix\_not\_del\_not\_fixed$$
$$pre(\varphi_0^-) = pre(\varphi^-) \cup \{\neg patched(a)\}$$
$$cost(\varphi_0^-) = C_0^-$$

- Actions $\varphi_1^-$: the atom to repair the action is not a negated effect, but the action has already been repaired before:

$$name(\varphi_1^-) = fix\_not\_del\_fixed$$
$$pre(\varphi_1^-) = pre(\varphi^-) \cup \{patched(a)\}$$
$$cost(\varphi_1^-) = C_1^-$$

- Actions $\varphi_0^+$: the atom to repair the action is a negated effect and the action has not been repaired before:

$$name(\varphi_0^+) = fix\_del\_not\_fixed$$
$$pre(\varphi_0^+) = pre(\varphi^+) \cup \{\neg patched(a)\}$$
$$cost(\varphi_0^-) = C_0^+$$

- Actions $\varphi_1^+$: the atom to repair the action is a negated effect and the action was repaired before:

$$name(\varphi_1^+) = fix\_del\_fixed$$
$$pre(\varphi_1^+) = pre(\varphi^+) \cup \{patched(a)\}$$
$$cost(\varphi_1^+) = C_1^+$$

This last action is the one that applies the least restrictive reparation, but has the highest cost. The different costs are distributed following $C_1^+ > C_0^+ > C_1^- > C_0^-$. By using the four defined types of fix actions with different costs, we want to prevent the exposed situations and promote the use of all available domain actions.

**Use of repaired actions.**   To control the application of repaired actions, we create two specific cases of close actions, $\omega^-$ and $\omega^+$. They are defined with identical scheme as $\omega$, but considering whether the action being closed is (or is not) a fixed action. Then, $w^-$ is defined as:

$$name(\omega^-) = close$$
$$pre(\omega^-) = pre(\omega) \cup \{\neg patched(a)\}$$
$$cost(\omega^-) = 0$$

And $w^+$ is defined as:

$$name(\omega^+) = close\_fix$$
$$pre(\omega^+) = pre(\omega) \cup \{patched(a)\} \cup \{fixed\}$$
$$cost(\omega^+) = C_\omega$$

where $C_\omega > 0$.

Therefore, we aim to minimize the cost of the reparations depending on where they are made, while also penalizing the application of repaired actions. With these diverse costs want to find more refined reparations to get as close as possible to the underlying domain, which corresponds to the user's mental model.

After considering the explained heuristic improvements, the resulting set of action schemes $\mathcal{A}'$ is the union of:

$$\mathcal{A}_\alpha = \{\alpha_a \mid a \in \mathcal{A}\}$$
$$\mathcal{A}_\varphi = \{\varphi_0^-, \varphi_1^-, \varphi_0^+, \varphi_1^+\}$$
$$\mathcal{A}_\psi = \{\psi_n, \psi_n^g \mid n\, arity\, of\, p(t) \in \mathcal{P}\}$$
$$\mathcal{A}_\omega = \{\omega^-, \omega^+\}$$

The following plan is part of a solution obtained for the extended planning task for the barman domain, where the fix and add-fix repair actions are highlighted:

```
...
12:(clean-shot shot01 ingredient01 left right)
13:(fix-not-del-not-fixed clean clean-shot)
14:(add-fix-1 clean clean-shot shot01 o)
15:(close-fixed clean-shot)
16:(fill-shot shot01 ingredient02 left right dispenser2)
17:(close-nofixed fill-shot)
18:(pour-shot-used-shaker shot01 ingredient02 shaker1 left l1 l2)
19:(close-nofixed pour-shot-to-used-shaker)
20:(clean-shot shot01 ingredient02 left right)
21:(add-fix-1 clean clean-shot shot01 o)
22:(close-fixed clean-shot)
...
```

It is important to note that if we perform the compilation on a completely specified task, the resulting plan will also achieve the goals, but without including any reparation.

## 6   Experiments

To empirically evaluate the feasibility of our approach, we selected the domains TRANSPORT, BLOCKSWORLD, ROVERS and BARMAN from the IPC. They were chosen to verify how the approach works as the number of actions and propositions increases. Table 1 summarises their main characteristics. We generated a set of 10 problems associated to each domain by increasing the number of objects involved, being the first problem the smaller one.[3] Each domain and problem conform to a complete planning task $\Pi_i$, being $i = \{1, \ldots, 10\}$ the number of the problem to which is associated. In order to test our approach, we created a set of add *add-incomplete* tasks with respect to each $\Pi_i$ by randomly deleting positive effects of the domain actions, provided that such changes made the task unsolvable. For instance, from the blocksworld domain we deleted the *holding* effect when it picks up a block or we removed a shot as *cleaned* from the *clean-shot* action for the barman domain.

We follow an iterative process in which we generate a set of tasks $\Pi_{i\mathcal{D}}^-$ where $\mathcal{D} = \{1, \ldots, 5\}$ is the number of removed positive effects with respect to $\Pi_i$. Therefore we have a total of 50 *add-incomplete* planning tasks for each domain.

---

[3] https://github.com/AI-Planning/pddl-generators

| Domain | $|\mathcal{A}|$ | $|\mathcal{P}|$ |
|---|---|---|
| TRANSPORT | 3 | 5 |
| BLOCKSWORLDS | 4 | 5 |
| ROVERS | 9 | 25 |
| BARMAN | 12 | 15 |

Table 1: Benchmark summary

They are compiled following the explained approach and solved to obtain the set of new effects $\hat{R}$ used in the reparation. We have tested the aforementioned planning tasks using the `seq-opt-lmcut` and `lama` configurations of Fast-Downward (Helmert 2006), using the LMCUT admissible heuristic (Helmert and Domshlak 2009) and an anytime planner that reports the best plan found in a given time window LAMA (Richter and Westphal 2010), respectively. The planning times for both configurations were set to 900 seconds.

The results are shown in Tables 2 and 3. We compared the set $\hat{R}$ obtained as solution with the actual removed that represents $R^*$, showing in $\%$ the percentage of the domain repaired and in $t$ the planning time in seconds. Empty fields mean that no plan has been found in the given time window. In general, the proposed approach performs well for smaller instances of domains and problems, although the scalability decreases in larger tasks. However, LAMA seems to be a better option since its use increases significantly the number of solved tasks with similar or shorter planning times. In most of the cases, we do not need to wait until the optimal solution of the planner to achieve the desired repair suggestions, intermediate plans already contains them. Rapid responses are also critical in the context of user support, and although some reparations exceed the repair rate (they include extra reparations besides the desired ones), domain designers can supervised the solutions suggested and discard unwanted reparations, even refining the search to obtains new plans which may show new suggestions.

## 7 Discussion

In this section we argue the proposed method along with the solutions provided. The main decision of this work relies on the applied method, which may have any other technique based on cost optimization as alternative. However, we implemented an AP compilation of the unsolvable task to keep the goal oriented approach, ensuring that the given solution also achieve the original goals of the problem. It provides a guideline for the reparations, showing those ones which help to achieve the goal.

Reparations are dependent on the problem configuration and the established metric provides an heuristic, presenting several ways to achieve the goals. To exemplified this, let us consider again the clean shot example. To achieve a clean glass it has to repair the clean operator, but the domain contains a very similar operator: empty-shot. This situation can return a solution in which the operator fixed is the latter. The effect is the same and the problem turns solvable any-

way, the only difference is semantic. But semantic issues are out of the scope of this work, delegating them to the users' side. By embedding the proposed system in any PDDL editor, users have the opportunity to decide if the proposed solution fits in its domain model or a better option it is to apply the reparation made to another operator. For larger domains and problems with multiple missing effects, were the performance of the method decreases, it is also possible to follow an iterative process. Instead of repair all the effects at once, the user can select some of the recommendations and run the system again, obtaining more precise suggestions since the model is now more accurate.

This approach also has the advantage of being parameterizable. For further developments, we can offer to users the possibility of choosing the penalizations applied to each operator, in addition to extend the metric with extra parameters as the plan length, for which it would be sufficient to also penalize the close operators.

## 8 Related Work

Several works addressed the previous problem in a effort to help users in the development of AP tasks. In *Mixed-initiative planning* (Burstein and McDermott 1996), planning is seen as a collaborative activity in which given a domain, a problem and a solution, automated and humans planners need to interact to jointly build a plan to accomplish a certain objective. Such solution can be generated from similar stored past plans (Veloso, Mulvehill, and Cox 1997) or generated by hand by the user (Howey and Long 2003; Howey, Long, and Fox 2004, 2005; Fox, Howey, and Long 2005), where if the plan is flawed it gives advice on how the plan should be fixed. Similarly, *Model reconciliation* (Chakraborti et al. 2017) presents way of bringing the human model closer to the agent's model by explaining the plan when it differs from the expected one.

These works rely on a properly specified domain and problem and also assume an initial plan which is changed or improved in collaboration with humans. However, there may be some cases where there is no suggested plan and the AP task proves to be unsolvable anyway. Techniques involved in such problems try to explain *why* it fails and *how* it could be solved. (Sreedharan et al. 2019) propose to assist the user by identifying unreachable subgoals of the problem. Since is challenging to extract subgoals from a unsolvable problem, they derive them from abstract and solvable models by using planning landmarks (Hoffmann, Porteous, and Sebastia 2004). Going a step further, given an unsolvable task it is also possible to make it solvable (Göbelbecker et al. 2010). Based on counterfactuals theory (Ginsberg 1985), it is able to explain why a plan fails and what should be done to prevent it, creating *excuse states* from which the given task is solvable, although they only consider changes in the initial state.

Planning with incomplete domains or approximate domain models (McCluskey, Richardson, and Simpson 2002; Simpson, Kitchin, and McCluskey 2007; Nguyen, Sreedharan, and Kambhampati 2017; Garland and Lesh 2002) considers not properly specified domains, but all those works assume that the incompleteness in the model is filled with

| Domain | $\mathcal{D}$ | $\Pi_{1\mathcal{D}}$ | | $\Pi_{2\mathcal{D}}$ | | $\Pi_{3\mathcal{D}}$ | | $\Pi_{4\mathcal{D}}$ | | $\Pi_{5\mathcal{D}}$ | | $\Pi_{6\mathcal{D}}$ | | $\Pi_{7\mathcal{D}}$ | | $\Pi_{8\mathcal{D}}$ | | $\Pi_{9\mathcal{D}}$ | | $\Pi_{10\mathcal{D}}$ | |
|---|---|---|---|---|---|---|---|---|---|---|---|---|---|---|---|---|---|---|---|---|---|
| | | % | t | % | t | % | t | % | t | % | t | % | t | % | t | % | t | p9 | t | p10 | t |
| TRANSPORT | 1 | 100 | 0,00 | 100 | 0,01 | 100 | 0,06 | 100 | 0,05 | 100 | 1,27 | 100 | 18,15 | 100 | 68,59 | - | - | - | - | - | - |
| | 2 | 100 | 0,05 | 100 | 0,24 | 100 | 0,74 | 100 | 0,07 | 100 | 5,80 | 100 | 12,91 | - | - | - | - | - | - | - | - |
| | 3 | 100 | 0,01 | 100 | 16,18 | 100 | 1,45 | 70 | 0,31 | - | - | - | - | - | - | - | - | - | - | - | - |
| | 4 | 100 | 0,20 | 100 | 21,25 | 100 | 2,55 | 100 | 96,01 | - | - | - | - | - | - | - | - | - | - | - | - |
| | 5 | 80 | 0,20 | 80 | 0,59 | - | - | 80 | 92,99 | - | - | - | - | - | - | - | - | - | - | - | - |
| BLOCKS WORLD | 1 | 100 | 0,01 | 100 | 0,01 | 100 | 0,01 | 100 | 0,03 | 100 | 0,01 | 100 | 0,11 | 100 | 0,06 | 100 | 0,03 | 100 | 0,32 | 100 | 3,07 |
| | 2 | 50 | 0,01 | 100 | 0,01 | 100 | 0,01 | 50 | 0,01 | 100 | 0,01 | 100 | 0,18 | 50 | 0,39 | 50 | 0,01 | 100 | 0,14 | 50 | 2,94 |
| | 3 | 70 | 0,00 | 70 | 0,05 | 70 | 0,15 | 100 | 0,02 | 35 | 0,00 | 70 | 1,01 | 70 | 1,69 | 35 | 3,71 | 70 | 114,52 | 100 | 32,30 |
| | 4 | 50 | 0,01 | 100 | 0,12 | 50 | 0,11 | 100 | 0,20 | 75 | 0,01 | 100 | 5,48 | 75 | 2,36 | 25 | 0,01 | 75 | 91,80 | - | - |
| | 5 | 80 | 0,01 | 100 | 0,03 | 80 | 0,23 | 60 | 0,66 | 80 | 0,03 | 100 | 16,80 | 60 | 385,66 | 60 | 0,13 | 100 | 34,76 | 40 | 798,06 |
| ROVERS | 1 | 100 | 48,00 | - | - | - | - | 100 | 418,63 | 100 | 697,40 | - | - | - | - | - | - | - | - | - | - |
| | 2 | 50 | 67,77 | - | - | - | - | - | - | 100 | 133,26 | - | - | - | - | - | - | - | - | - | - |
| | 3 | 35 | 808,46 | - | - | - | - | 65 | 560,42 | 65 | 275,03 | - | - | - | - | - | - | - | - | - | - |
| | 4 | 25 | 205,27 | - | - | - | - | - | - | - | - | - | - | - | - | - | - | - | - | - | - |
| | 5 | 40 | 709,13 | - | - | - | - | 40 | 348,30 | - | - | - | - | - | - | - | - | - | - | - | - |
| BARMAN | 1 | 100 | 1,75 | 0 | 11,99 | 100 | 3,97 | 100 | 725,80 | 0 | 807,99 | - | - | - | - | - | - | - | - | - | - |
| | 2 | 50 | 16,87 | 50 | 11,71 | 100 | 27,60 | - | - | - | - | - | - | - | - | - | - | - | - | - | - |
| | 3 | - | - | - | - | 65 | 589,28 | - | - | - | - | - | - | - | - | - | - | - | - | - | - |
| | 4 | - | - | - | - | - | - | - | - | - | - | - | - | - | - | - | - | - | - | - | - |
| | 5 | - | - | 40 | 550,38 | 40 | 387,40 | - | - | - | - | - | - | - | - | - | - | - | - | - | - |

Table 2: Results of the extended planning tasks using the LMCUT configuration of Fast-Downward. Parameters shown the initial number of removed effects ($\mathcal{D}$), the percentage (%) of reparation achieved compared to the initial number of removed effects and the planning time ($t$) spent to find the solution.

| Domain | $\mathcal{D}$ | $\Pi_{1\mathcal{D}}$ | | $\Pi_{2\mathcal{D}}$ | | $\Pi_{3\mathcal{D}}$ | | $\Pi_{4\mathcal{D}}$ | | $\Pi_{5\mathcal{D}}$ | | $\Pi_{6\mathcal{D}}$ | | $\Pi_{7\mathcal{D}}$ | | $\Pi_{8\mathcal{D}}$ | | $\Pi_{9\mathcal{D}}$ | | $\Pi_{10\mathcal{D}}$ | |
|---|---|---|---|---|---|---|---|---|---|---|---|---|---|---|---|---|---|---|---|---|---|
| | | % | t | % | t | % | t | % | t | % | t | % | t | % | t | % | t | % | t | % | t |
| TRANSPORT | 1 | 100 | 0,01 | 100 | 0,02 | 100 | 0,03 | 100 | 0,03 | 100 | 1,04 | 100 | 0,08 | - | - | 100 | 197,55 | - | - | - | - |
| | 2 | 100 | 0,01 | 100 | 0,01 | 150 | 0,08 | 100 | 17,72 | 100 | 5,17 | 100 | 47,40 | 150 | 63,53 | - | - | - | - | - | - |
| | 3 | 100 | 0,10 | 100 | 0,02 | 100 | 178,80 | 70 | 51,35 | 100 | 1,93 | 100 | 1,51 | - | - | - | - | - | - | - | - |
| | 4 | 100 | 0,18 | 100 | 0,24 | 100 | 0,21 | 100 | 210,14 | 100 | 6,69 | 125 | 80,85 | - | - | - | - | - | - | - | - |
| | 5 | 80 | 0,19 | 80 | 0,23 | 100 | 0,19 | 120 | 37,47 | 100 | 6,64 | 120 | 77,09 | - | - | - | - | - | - | - | - |
| BLOCKS WORLD | 1 | 100 | 0,00 | 100 | 0,01 | 100 | 0,01 | 100 | 0,01 | 100 | 0,01 | 100 | 0,03 | 100 | 0,01 | 100 | 0,01 | 100 | 0,04 | 100 | 0,03 |
| | 2 | 100 | 0,00 | 100 | 0,17 | 100 | 0,01 | 100 | 0,01 | 100 | 0,01 | 100 | 0,22 | 100 | 0,01 | 50 | 0,01 | 100 | 0,04 | 50 | 0,25 |
| | 3 | 70 | 0,01 | 70 | 0,01 | 70 | 0,02 | 100 | 0,01 | 35 | 0,01 | 70 | 0,21 | 70 | 0,36 | 70 | 0,13 | 70 | 0,04 | 100 | 0,27 |
| | 4 | 75 | 0,02 | 100 | 0,18 | 75 | 0,04 | - | - | 75 | 0,51 | 100 | 16,89 | 75 | 21,95 | 75 | 0,02 | 75 | 0,04 | 75 | 0,38 |
| | 5 | 100 | 0,01 | 100 | 0,22 | 100 | 0,22 | 60 | 0,02 | 100 | 0,05 | 100 | 2,74 | 80 | 0,58 | 80 | 0,10 | 100 | 6,86 | 80 | 16,96 |
| ROVERS | 1 | 0 | 0,03 | 200 | 0,03 | 200 | 0,04 | - | - | 200 | 0,05 | - | - | 0 | 1,28 | - | - | 200 | 0,28 | 300 | 31,36 |
| | 2 | 50 | 0,03 | 50 | 0,08 | 150 | 59,28 | - | - | 150 | 0,38 | - | - | 0 | 0,21 | 250 | 2,83 | 50 | 0,27 | - | - |
| | 3 | 0 | 50,98 | 70 | 0,03 | 70 | 0,75 | 70 | 1,33 | 35 | 1,08 | - | - | - | - | 35 | 0,30 | 0 | 1,47 | 35 | 0,73 |
| | 4 | - | - | 50 | 0,09 | 25 | 0,16 | 50 | 0,49 | 50 | 16,82 | - | - | - | - | 40 | 0,19 | - | - | - | - |
| | 5 | 80 | 1,03 | 80 | 0,06 | - | - | 20 | 0,45 | 40 | 12,85 | - | - | - | - | - | - | - | - | - | - |
| BARMAN | 1 | 0 | 0,10 | 200 | 0,06 | 200 | 0,10 | - | - | 100 | 264,45 | 100 | 447,26 | - | - | - | - | - | - | 200 | 1,30 |
| | 2 | 50 | 0,24 | 50 | 3,27 | 50 | 0,13 | - | - | - | - | 250 | 36,40 | - | - | - | - | - | - | - | - |
| | 3 | - | - | 35 | 0,08 | - | - | - | - | - | - | - | - | - | - | - | - | - | - | - | - |
| | 4 | - | - | 50 | 3,50 | 50 | 4,22 | - | - | - | - | - | - | 25 | 10,87 | - | - | - | - | - | - |
| | 5 | - | - | 40 | 3,59 | 60 | 0,93 | - | - | - | - | 40 | 4,76 | - | - | - | - | 80 | 27,79 | - | - |

Table 3: Results of the extended planning tasks using the LAMA configuration of Fast-Downward. Parameters shown the initial number of removed effects ($\mathcal{D}$), the percentage (%) of reparation achieved compared to the initial number of removed effects and the planning time ($t$) spent to find the solution.

annotations or statements about where the model has been incompletely specified, and providing guidelines to supply the lack of knowledge.

Therefore, all of the works seen so far make crucial assumptions about the domain engineer understanding of the problem or do not deal with partially specified domains without further help of the user. In this paper we work on fixing domains that have some missing actions' effects which make the problem unsolvable, and we have no *a priory* clues about where the error resides.

## 9 Conclusions

We have presented a novel approach for the reparation of unsolvable planning tasks due to a missing positive effects in its domain, which is based on classical planning techniques. Since the given task is incomplete, we have proposed its compilation to a new extended planning task which intro-duces general operators to repair any of the actions of the domain by linking them to new positive effects. To test our method, we selected some domains from the IPC to generate complete planning tasks and altering its domain by randomly deleting positive effects, provided that such changes turn the task unsolvable. The resulting incomplete task is compiled to the new extended task and solved by an optimal planner, including in the plan the modification on the actions made while also achieving the original goals. Results show that the approach performs well for a wide range of planning tasks, especially with an anytime planner configuration. We believe that this work has enough potential to be considered as a user support system to develop planning tasks.

## 10 Acknowledgements

This work has been partially funded by the EU ECHORD++ (FP7-ICT-601116) project, FEDER/Ministerio de Ciencia

e Innovación - Agencia Estatal de Investigación RTI2018-099522-B-C43 project. It has been also supported by the Madrid Government (Comunidad de Madrid, Spain) under the Multiannual Agreement with UC3M in the line of Excellence of University Professors (EPUC3M17) in the context of the V PRICIT (Regional Programme of Research and Technological Innovation).

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
