# OpenReview forum: "Repair Suggestions for Planning Domains with Missing Actions Effects"
_icaps-conference.org/ICAPS/2022/Workshop/XAIP — XAIP 2022_

### Official Review · Reviewer_xJw6 · 2022-04-28
**Review for Paper #11**

**Rating:** 7
**Confidence:** 4

**Review:**

# Summary

The authors present a novel approach for fixing and repairing incomplete (deterministic) planning tasks with missing (positive) effects.
Such approach is essentially  a compilation approach that generates a new extended incomplete planning task with new operators that are able to to repair the incomplete actions by associating such incomplete actions to new (correct) positive effects. Experiments over four distinct IPC domains models and hundreds incomplete planning tasks show the feasibility and scalability of the proposed approach for repairing  and fixing incomplete planning tasks.

# Detailed Comments

1. I am not sure if $\mathcal{M}^{*}\_{\Pi}$ should be called as an **optimal** solution to $\Pi^{-}$.
I would say that $\mathcal{M}^{*}\_{\Pi}$ should be called as an optimal solution to x I would say that $\mathcal{M}_{\Pi}^{*}$ is the correct (exact) set of missing effects that makes the task $\Pi^{-}$ solvable, and consequently $\Pi = \Pi^{-}$. Please, consider revising and renaming Definition 3.

2. You formally define an action $a \in A$ as a tuple {pre(a), add(a), del(a)}. However, in Section 4.2, you define a repair action using $eff$. Since a repair action only has positive effects, you should use $add$, just to be consistent with what you defined in Section 2.

3. The authors do not cite some important papers related to incomplete (STRIPS) domain models, such as:
- Planning and Acting in Incomplete Domains, Weber and Bryce, ICAPS, 2011;
- Reactive, Proactive, and Passive Learning about Incomplete Actions, Weber and Bryce, ICAPS Workshop on Planning and Learning, 2011;
The second paper is actually very related to your work, so please try to add comment on that by contrasting your work with Weber and Bryce's paper.

4. As for your experiments, you could have used more modern and recent (optimal, safisticing, and agile) planners, so you would be able solve many more tasks (https://ipc2018-classical.bitbucket.io/#planners).

5. Typos and writing issues:
- "Specially when the domain is tough to represent.". I would say "difficult to represent";
- "Completeness and soundness of the compilation is discussed in Section 5" -> "The completeness and soundness of the compilation are discussed in Section 5";
- "The solution consist in" -> "The solution consists in";
- "... task with respecto ..." -> "... task with respect to ...";
- "Each domain and problem conform a complete planning task" -> "Each domain and problem conform to a complete planning task";
- "The results are shown in tables Figure 2 and 3". I guess you mean Tables 2 and 3, right?! Please, fix that.

# Questions

Please, try to address the following questions in the next version of the paper (or in the camera-ready version of the paper, in case of acceptance).

1. How difficult would be to adapt your compilation approach to also fix domains models with missing preconditions?

2. You say "Since at least one action lacks some positive effect(s), the planning task is extended with general operators to repair them.". That is a strong assumption.
What if there is no action with missing positive effects to repair/fix? Please, justify such assumption.

# Overall Evaluation and Decision

The paper is overall well-written, organized, and easy to follow. I think the paper is relevant to the workshop's topic, so I recommend the acceptance of the paper.

---

### Official Review · Reviewer_XazH · 2022-04-30
**An acceptable paper**

**Rating:** 7
**Confidence:** 3

**Review:**

The paper intends to turn an unsolvable planning problem into a solvable one by adding state variables to actions' positive effects, and such changes can serve as an explanation about why the problem is unsolvable. The author solved the problem of finding such changes by reducing it to another planning problem. The reduction proposed by the authors is sound and complete, and hence, it can serve the goal of finding the desired changes. Generally speaking, I think it is a good paper and can be accepted.

**Significance**: I think the paper is a good extension of the one by Gobelbecker et al. (2010) in which the authors only considered modifying the initial state of a planning problem. Although the extension is straightforward, the problem it intends to solve can happen frequently in practice, because for a non-expert planning problem modeler, it is quite often the case that he/she would make some mistakes in modeling actions.

Since the authors want to change actions' effects (more concretely, positive effects), I wonder whether the authors have considered changing actions' preconditions and negative effects as well? I think those two changes are also important because the unsolvability of a planning problem might also be caused by some action deleting a state variable or some action's precondition not being able to be satisfied (though the latter one can be fixed via changing actions' positive effects as well).

**Clarity**: The paper is easy to follow in general. However, I also found some inconsistencies in the paper. More concretely, the authors aim to change actions' positive effects in a *lifted* planning problem, but the planning formalism presented in the paper is for *grounded* planning problems. Further, the authors did not mention *negative* preconditions in the definition of the planning formalism, but when constructing some actions in a new planning problem, the negative precondition $\neg \mathit{open}$ is involved. Closely related, I also want to ask how will the reduction (construction of the new planning problem) change when negative preconditions are taken into account. Further, I also want to ask that in the empirical evaluation, is it possible that some state variables are added to some actions' positive effects which are not supposed to be there? If such a situation can happen, how will it affect the calculation of the percentage of reparation?

**Scholarship**: The authors have covered sufficient related works in the paper.

**Reproducibility**: I think that is good.

**Soundness**: I did not find any significant error.

**Minor comment(s)**: Using \mathit{eff} instead of eff in the math environment.

---

### Meta-Review · Program_Chairs · 2022-04-30

**Recommendation:** Accept
**Confidence:** 5

**Metareview:**

The papers looks at a compilation approach for repairing classical planning problems with missing effects. This is a good approach for explaining unsolvable planning tasks and we would like to see a discussion and presentation in the workshop. As you move forward, please consider the reviewers’ feedback, particularly if the proposed method can be extended to missing preconditions and negative effects. This would make your approach much stronger.

On another note, we want to emphasize on the lack of explainability in the paper. While the proposed method can be seen as “explaining” why a planning problem cannot be solved by (loosely speaking) counterfactual reasoning, i.e., the problem can be solved if this operator is included, the notion of explainability is not explicit in the paper. As the main theme of the workshop is on explainable planning, we suggest the authors make the connection to explainability apparent. For example, how would an explanation look like? How would you communicate the reparations (e.g., explanations) to human users and so on.

We are looking forward to your presentation.

---

### Decision · Program_Chairs · 2022-04-30

Accept